# Atypical Associations between Functional Connectivity during Pragmatic and Semantic Language Processing and Cognitive Abilities in Children with Autism

**DOI:** 10.3390/brainsci13101448

**Published:** 2023-10-11

**Authors:** Amparo V. Márquez-García, Bonnie K. Ng, Grace Iarocci, Sylvain Moreno, Vasily A. Vakorin, Sam M. Doesburg

**Affiliations:** 1Department of Biomedical Physiology and Kinesiology, Simon Fraser University, Burnaby, BC V5A 1S6, Canada; 2School of Interactive Arts and Technology, Simon Fraser University, Surrey, BC V3T 0A3, Canada; 3Department of Psychology, Simon Fraser University, Burnaby, BC V5A 1S6, Canada; grace_iarocci@sfu.ca

**Keywords:** autism, pragmatic language, semantic language, fMRI, functional connectivity, brain–behavior associations, network neuroscience

## Abstract

Autism Spectrum Disorder (ASD) is characterized by both atypical functional brain connectivity and cognitive challenges across multiple cognitive domains. The relationship between task-dependent brain connectivity and cognitive abilities, however, remains poorly understood. In this study, children with ASD and their typically developing (TD) peers engaged in semantic and pragmatic language tasks while their task-dependent brain connectivity was mapped and compared. A multivariate statistical approach revealed associations between connectivity and psychometric assessments of relevant cognitive abilities. While both groups exhibited brain–behavior correlations, the nature of these associations diverged, particularly in the directionality of overall correlations across various psychometric categories. Specifically, greater disparities in functional connectivity between the groups were linked to larger differences in Autism Questionnaire, BRIEF, MSCS, and SRS-2 scores but smaller differences in WASI, pragmatic language, and Theory of Mind scores. Our findings suggest that children with ASD utilize distinct neural communication patterns for language processing. Although networks recruited by children with ASD may appear less efficient than those typically engaged, they could serve as compensatory mechanisms for potential disruptions in conventional brain networks.

## 1. Introduction

Autism spectrum disorder (ASD) is a heterogeneous neurodevelopmental condition with a current prevalence of 1 in 36 [1]. ASD can be diagnosed in children as early as 2 years of age [2,3]. Social communication difficulties are a fundamental trait of individuals with ASD [4,5]. Specifically, individuals with ASD often struggle with utilizing social and contextual cues in language processing [6,7]. Additionally, children with ASD are commonly faced with challenges in social interactions and communication and display repetitive or restricted behaviors [8,9], all of which can impact their learning and their ability to establish connections with the world. However, our current understanding of the intricate relationship between brain activity, functional connectivity, and abilities in these areas and other related domains remains limited.

Considerable effort has been made to investigate the manifestation of autistic traits and how they differ from neurotypical individuals [2,10,11,12]. When it comes to understanding atypical brain development in individuals with ASD, despite some findings from post-mortem studies [13] and some invasive approaches such as blood analyses [2,14], neuroimaging-related studies have continued to be a mainstay of knowledge generation regarding brain–behavior relations in this group. One of the most prolific approaches for studying ASD using neuroimaging is to investigate functional connectivity through functional magnetic resonance imaging (fMRI) [12]. Functional connectivity, as studied using fMRI, refers to the statistical correlations between oxygen levels in various regions of the brain [15,16]. It is believed that there is communication between the brain regions that express hemodynamic correlations [15]. Researchers have reached a consensus that individuals with ASD exhibit atypical patterns of functional brain connectivity [12,17]; however, the specific ways in which individuals with ASD differ from their neurotypical peers are still an active area of research.

Research has consistently shown that individuals with ASD can exhibit both increased and decreased connectivity compared with typically developing (TD) individuals, depending on factors such as the individual participants, age range, analysis methods, whether the connectivity is local or long-range, as well as the specific brain regions under investigation [11,12,17]. One commonly reported finding is a reduced long-range within-network connectivity, for example, between the prefrontal cortex (PFC) and the posterior regions of the brain in individuals with ASD [11]. Additionally, reduced functional lateralization has been observed in ASD compared with TD [18], with greater activation in the right hemisphere during language tasks [19]. Although less frequently reported, a reduction in long-range connectivity during resting-state has also been observed [20], while other studies suggest that local connectivity between different networks may be either increased or decreased in ASD compared with typically developing individuals [11,12]. 

Regarding the relationship between functional connectivity and structural connectivity, it seems that the reduced long-range within-network connectivity in ASD is associated with lower network differentiation in ASD compared with TD [17], which is reflected in reduced synaptic stabilization and axonal myelination [21,22]. Conversely, increased local connectivity is associated with processes such as synaptic pruning [17]. Such findings suggest that both under and over-connectivity are present in an ASD brain and that they may reflect different but equally important aspects of network function and/or dysfunction. 

The use of neuroimaging modalities to compare individuals with ASD and typically TD individuals has also provided valuable insights into the pragmatic language processing networks [23,24,25,26,27,28]. These studies have revealed aberrant functional connectivity between cortical areas during a range of language tasks, suggesting that alterations in cortical connectivity and deficient communication among cortical regions may contribute to the language difficulties observed in ASD [29,30]. Moreover, evidence has linked symptoms of ASD to atypical connectivity and neural activation within the salience network and the default mode network (DMN). Specifically, studies have demonstrated a dysfunctional anterior insula and abnormal activity in the anterior cingulate cortex in ASD, both of which are components of the salience network [31]. The anterior cingulate cortex, known for its role in social decision-making, processing other-oriented information, and tracking the motivation of others, is particularly relevant in understanding ASD [31]. The Inferior Frontal Gyrus (IFG), a region associated with speech and language processing, working memory, empathy processing, sarcasm detection, and inhibition, also plays a crucial role in pragmatic language processing [32,33,34]. Additionally, research suggests that children with ASD activate alternative and possibly less flexible networks during phonetic, semantic, syntactic, and pragmatic language processing [35]. Taken together, the growing body of research on altered connectivity among brain regions in ASD provides support for the hypothesis that social and pragmatic language difficulties in individuals with ASD arise from atypical connectivity within and between brain networks.

There has been abundant research on atypical functional connectivity in ASD [36]. Additional studies are needed, however, to better illuminate how connectivity alterations in ASD are related to cognitive abilities. It also remains unclear whether and to what extent these atypical connectivity patterns are compensatory or maladaptive in ASD. To address this, the present study aimed to characterize associations between task-dependent functional connectivity during semantic and pragmatic language processing and cognitive abilities in relevant domains in children with ASD and their typically developing peers. To our knowledge, this is the first study to investigate brain–behavior relationships in ASD in relation to TD in the context of pragmatic and semantic language processing. 

There are two main goals of this study. First, we aimed to understand the relationship between task-dependent functional connectivity during pragmatic and semantic language processing and cognitive abilities in children with ASD and their TD peers. To study the hypothesis that task-dependent connectivity in ASD and TD relates to their cognitive performance, we performed behavioral partial least squares (PLS) [37,38] by analyzing fMRI and psychometric data for ASD and TD groups separately using a data augmentation approach. The results are indicative of the brain–behavior relationship within each group. Second, we endeavored to understand the relationship between differences observed in ASD and TD groups in terms of brain–behavior relationships. Unlike our first set of analyses performed separately on the ASD and TD groups, this second set of analyses is a direct comparison between the ASD and TD groups to understand their differences. To test the hypothesis that the ASD group has different brain–behavior correlations compared with the TD group, we computed the differences between ASD and TD in both fMRI data and behavioral information gathered from psychometric assessments. We then investigated correlations in differences in fMRI and differences in behavioral data. This is a direct test of whether the differences in connectivity between the groups are related to their differences in behavior. 

## 2. Methods

### 2.1. Participants

This study involved two groups of children: TD and those diagnosed with ASD. The ASD group comprised 16 children aged between 7.2 and 12 years old (3 females, 13 males), with an average age of 10.15 years old (SD = 1.32). These children had received a diagnosis of ASD. The control group comprised 19 children who did not have ASD or other neurodevelopmental disorders (6 females, 13 males), with an average age of 9.55 years old (SD = 1.56), ranging from 7 to 12 years old. Children with standard contraindications for Magnetic Resonance Imaging (MRI), such as metallic implants, were excluded from the study. After pre-processing and removing missing data, we retained for analysis 11 ASD participants (2 females, 9 males; M = 9.85 years old, SD = 1.19 years old) and 16 TD participants (6 females, 10 males; M = 9.6375 years old, SD = 1.59 years old). The sample statistics are outlined in Table 1 below. This study was approved by the Simon Fraser University Research Ethics Board (#20150258, 6 October 2021).

Prior to undergoing fMRI scanning, the participants’ cognitive abilities were evaluated. The assessment revealed that none of the participants had intellectual disabilities. When comparing the groups based on the Full-Scale Intelligence Quotient 2 (FSIQ 2) from the WASI assessment (specifically, vocabulary and matrix reasoning subsets), there was no statistically significant difference between the groups (ASD: mean = 102.75, SD = 19.81; TD: M = 110.84, SD = 11.44; *p* = 0.16). When the participants completed the Autism Quotient (AQ) questionnaire [2,39], there was a notable discrepancy between the groups (ASD [n = 16, no missing data]: M = 32.63, SD = 7.83; Controls [*n* = 19, no missing data]: *M* = 13, *SD* = 6.94; *p* = *p* < 0.011). These results confirmed that individuals in the ASD group displayed more characteristics associated with autism. The results of the AQ questionnaire and FSIQ 2 are also outlined in Table 1 below. 

### 2.2. Behavioral Assessment

Each participant performed an approximately 1.5-h battery of neuropsychological tests to measure performance in language, restricted/repetitive behavior/interests, social cognition, attention, and executive function. The list of behavioral assessments is listed in Table 2. We also administered a neuropsychological battery which included a demographic Questionnaire, Social Style Questionnaire (Age 4–12), Chandler Cartoons Questionnaire (Sandcastle, Airplane, and Snowman) [40], Multidimensional Social Competence Scale (MSCS) [41], Behavior Rating Inventory of Executive Function: Third Edition (BRIEF-2) [42], Social Responsiveness Scale (SRS 2) [43], Comprehensive Assessment of Spoken Language (CASL 2) [44]; Wechsler Abbreviated Scale of Intelligence (WASI 2) [45] and, Behavior Assessment System for Children: Parent Rating Scales Report (BASC 3) [46]. 

### 2.3. Task Protocol

Evidence has indicated that language and social communication are areas in which children with ASD have difficulties [5,6,7]; we recorded brain activity from children while they watched videos with either semantic or pragmatic language components. We used two sets of nine video clips, totaling 18 clips, which were divided into two separate runs. Half of the videos contained a semantic component, while the other half had a pragmatic component. The details of the stimulus procedures are also described in Márquez-García et al. [47]. 

Each video depicted two individuals (referred to as the “Partner” and the “Speaker”) sitting across from each other at a table with eight objects placed on it (Figure 1). The video sequence began with the Partner uttering a context sentence, such as “What are these called?” or “What can I get you?” which determined the speech act in which the Speaker would pronounce the critical word. Following the context sentence, six of the eight objects on the table were named. Depending on the context sentence, the critical word was expressed as either a Naming or a Request speech act. Previous studies have utilized these two conditions to explore pragmatic and semantic language processing. Naming objects has been associated with semantic language processing, while performing an action has been linked to pragmatic language processing [48,49]. Each video had a duration of 20 s (Figure 1). A male and a female speaker were involved in the video recordings, with both individuals taking turns as the Partner and the Speaker in half of the videos. The left-right positions were also counterbalanced, with each position used in half of the videos. Apple’s iMovie 10.1 software was employed for video editing, and the complete blocks of videos were presented using Presentation Software. The same critical words and objects were used in both sets of videos. In order to ensure participant attention, a word was presented at the end of each block, and participants were asked if they had heard that word during the video. 

### 2.4. MRI Data Acquisition

The experimental procedure included functional and anatomical structural magnetic resonance imaging (MRI) scans. These MRI data were collected at ImageTech, a specialized imaging facility embedded into a public hospital in the Greater Vancouver area and affiliated with Simon Fraser University (SFU). ImageTech is equipped with research-focused 3T MRI capabilities and offers resources for psychometric assessment.

The sequence of assessments was adaptable and contingent on scanner availability. To maintain the children’s focus and motivation throughout the procedure, we provided refreshments such as snacks and juice, along with small incentives such as toys or movie passes. Assent was secured from the children, while their parents provided informed consent.

The imaging protocol was designed to include the acquisition of high-resolution T1-weighted (T1w) sagittal 3D MPRAGE images, providing anatomical information. In addition to this structural imaging, two runs of blood–oxygen-level-dependent (BOLD) functional MRI (fMRI) were conducted for each participant. During these runs, the subjects engaged in both pragmatic and semantic language tasks. The specific design and nature of these tasks were described above in the Stimulus Protocol section.

More specifically, the MRI scans were acquired using a 3 Tesla Philips Ingenia CX MRI scanner equipped with a 32-channel dStream head coil. For the functional MRI (fMRI) images, an echo planar imaging gradient-echo (GRE-EPI) sequence was applied, with specific acquisition parameters: repetition time (TR) of 2000 ms, echo time (TE) of 30 ms, and a flip angle of 90°. The in-plane resolution was configured at 3 × 3 mm, and the slice thickness was 3 mm, resulting in matrix dimensions of 80 × 80 voxels across 36 slices.

Complementing the functional scans, high-resolution T1-weighted (T1w) images were also collected during each session to enable precise co-registration with the functional images. The acquisition parameters for these T1w images were selected as follows: a TE of 3.7 ms, a flip angle of 8°, and a field of view (FOV) of 256 × 242. The slice thickness was set at 1 mm, and the images were captured with matrix dimensions of 256 × 242 voxels across 213 slices in the sagittal orientation.

### 2.5. fMRI Pre-Processing

To pre-process these MRI data, we applied a fMRIprep pipeline that uses a combination of packages such as FSL, ANTSs, FreeSurfer, and AFNI [50,51,52,53,54,55,56,57,58,59,60,61,62,63]. This pipeline was designed to provide automatic workflow for pre-processing fMRI data and co-register it with the Montreal Neurological Institute (MNI) space. It performs basic processing steps (co-registration, normalization, unwarping, noise component extraction, segmentation, and skull stripping). To detail the MRI data pre-processing steps, we incorporated the custom language generated by the fMRIprep software, as recommended by its developers. This inclusion ensures a comprehensive and standardized description within the main text.

Using fMRIPrep (version 20.2.1), we converted fMRI and T1w MRI data to Brain Imaging Data Structure (BIDS) format. The T1w images were corrected for intensity non-uniformity (INU) with N4BiasFieldCorrection [64], distributed with ANTs 2.3.3 [65] (RRID:SCR_004757), and used as T1w-reference throughout the workflow. The T1w-reference was then skull-stripped with a Nipype implementation of the antsBrainExtraction.sh workflow (from ANTs), using OASIS30ANTs as the target template. Brain tissue segmentation of cerebrospinal fluid (CSF), white–matter (WM), and gray–matter (GM) was performed on the brain-extracted T1w using fast (FSL 5.0.9, RRID:SCR_002823) [66]. Volume-based spatial normalization to one standard space (MNI152NLin2009cAsym) was performed through nonlinear registration with antsRegistration (ANTs 2.3.3), using brain-extracted versions of both T1w reference and the T1w template. The following template was selected for spatial normalization: ICBM 152 Nonlinear Asymmetrical template version 2009c (RRID:SCR_008796; TemplateFlow ID: MNI152NLin2009cAsym) [67].

For each of the 2 BOLD runs per subject (across all tasks and sessions), the following pre-processing was performed. First, a reference volume and its skull-stripped version were generated using fMRIPrep. Susceptibility distortion correction (SDC) was omitted. The BOLD reference was then co-registered with the T1w reference using flirt (FSL 5.0.9) [68] with the boundary-based registration cost-function [69]. Co-registration was configured with nine degrees of freedom to account for distortions remaining in the BOLD reference. Head-motion parameters with respect to the BOLD reference (transformation matrices and six corresponding rotation and translation parameters) were estimated prior to spatiotemporal filtering using mcflirt (FSL 5.0.9) [70]. BOLD runs were slice-time corrected using 3dTshift from AFNI, release 20160207 (RRID:SCR_005927) [71]. The BOLD time-series (including slice-timing correction when applied) were resampled onto their original, native space by applying the transforms to correct for head-motion. These resampled BOLD time series will be referred to as pre-processed BOLD in the original space or just pre-processed BOLD. The BOLD time-series were resampled into standard space, generating a pre-processed BOLD run in MNI152NLin2009cAsym space. First, a reference volume and its skull-stripped version were generated using a custom methodology of fMRIPrep. Several confounding time-series were calculated based on the pre-processed BOLD: framewise displacement (FD), DVARS, and three region-wise global signals. FD was computed using two formulations following Power (absolute sum of relative motions, Power et al. (2014) [72]) and Jenkinson (relative root mean square displacement between affines [70]. FD and DVARS were calculated for each functional run, both using their implementations in Nipype (following the definitions by Power et al. 2014) [72]. 

The three global signals were extracted within the CSF, the WM, and the whole-brain masks. Additionally, a set of physiological regressors was extracted to allow for component-based noise correction (CompCor) [73]. Principal components are estimated after high-pass filtering the pre-processed BOLD time-series (using a discrete cosine filter with 128s cut-off) for the two CompCor variants: temporal (tCompCor) and anatomical (aCompCor), for tCompCor components were then calculated from the top 2% variable voxels within the brain mask. For aCompCor, three probabilistic masks (CSF, WM, and combined CSF+WM) were generated in anatomical space. The implementation differed from that of Behzadi et al. in that instead of eroding the masks by 2 pixels on BOLD space, and the aCompCor masks subtracted a mask of pixels that likely contained a volume fraction of GM. This mask was obtained by thresholding the corresponding partial volume map at 0.05, and it ensured components were not extracted from voxels containing a minimal fraction of GM. Finally, these masks were resampled into BOLD space and binarized by thresholding at 0.99 (as in the original implementation). Components were also calculated separately within the WM and CSF masks. For each CompCor decomposition, the k components with the largest singular values were retained, such that the retained components’ time series was sufficient to explain 50 percent of variance across the nuisance mask (CSF, WM, combined, or temporal). The remaining components were dropped from consideration. The head-motion estimates calculated in the correction step were also placed within the corresponding confounds file. The confound time series derived from head motion estimates and global signals were expanded with the inclusion of temporal derivatives and quadratic terms for each [74]. Frames that exceeded a threshold of 0.5 mm FD or 1.5 standardized DVARS were annotated as motion outliers. All resamplings were performed with a single interpolation step by composing all the pertinent transformations (i.e., head-motion transform matrices, susceptibility distortion correction when available, and co-registrations to anatomical and output spaces). Gridded (volumetric) resamplings were performed using antsApplyTransforms (ANTs), configured with Lanczos interpolation to minimize the smoothing effects of other kernels [75]. Non-gridded (surface) resamplings were performed using mri_vol2surf (FreeSurfer). 

### 2.6. Estimation of Functional Connectivity during Pragmatic and Semantic Processing

Each participant’s pre-processed functional MRI data were further analyzed to characterize functional brain connectivity within the cerebral cortex. To this end, we applied the Schaefer atlas, a functional parcellation of the cortex that divides it into 500 distinct regions of interest (ROIs), each assigned to one of 17 resting-state networks. This atlas builds on the network architecture defined by Yeo et al. 2011 [76], who identified several distinct functional networks in the brain through clustering analysis of resting-state fMRI data. The networks were defined separately for the right and left hemispheres, resulting in 34 networks in total.

For each ROI, we extracted the characteristic time series by computing the mean fMRI time series, averaged across voxels within the given ROI. This procedure was performed separately for each run. Then, for each pair of ROIs in the parcellation, we computed three estimates of coordinated activity for each condition: baseline, semantic, and pragmatic. This was necessary to account for different numbers of data points for each condition, which could bias the estimates of connectivity. For example, in run 1, the baseline, semantic, and pragmatic conditions had 66, 55, and 55 data points, respectively, while in run 2, these numbers were 54, 44, and 44.

To mitigate this bias, we randomly selected 30 data points for each run, condition, and pair of ROIs and then computed distance correlation as a measure of connectivity between these two time series. This procedure was repeated 100 times, and the estimates of distance correlation across both runs were combined and median-calculated. This measure served as an estimate of functional brain connectivity for a given pair of brain regions. As a result of this procedure, each subject was associated with three condition-specific 500 × 500 matrices of functional brain connectivity, corresponding to the semantic and pragmatic conditions, as well as the baseline.

We then performed a baseline correction by subtracting these data during the baseline condition from the pragmatic and semantic conditions, respectively. This subtraction yielded two baseline-corrected 500 × 500 matrices corresponding to the pragmatic and semantic conditions. These connectivity matrices were subsequently correlated with psychometric measures, as described in the following section.

### 2.7. Associations between Brain Connectivity and Psychometrics

The primary question we aimed to address involves the statistical characterization of group differences between ASD and TD individuals in terms of correlations between psychometrics outcome measures and functional brain connectivity during a language task. Our procedure was as follows. First, we paired subjects from the TD group (*n* = 16) and the ASD group (*n* = 11) and computed the differences in both connectivity matrices and vectors of psychometric scores for each pair. This resulted in a total of 176 observations (11 × 16). We then correlated the differences in connectivity with the corresponding differences in psychometrics across these observations, using a multivariate analysis, as described below.

To establish a baseline for the results observed in the group differences, we applied a similar procedure separately for the TD and ASD groups. Specifically, within each group, we selected pairs of subjects and computed the differences in connectivity and the corresponding differences in psychometrics. This step was repeated for all possible pairs of subjects within each group, yielding 120 observations for the TD group (16 × 16 − 1)/2) and 55 observations for the ASD group (11 × (11 − 1)/2).

### 2.8. Multivariate Framework for Analysis of Correlations 

To explore and quantify the correlations between brain connectivity and psychometric scores and to assess their significance, we employed a multivariate statistical technique known as Behavioral Partial Least Squares (PLS) [37,38]. This PLS analysis was conducted six times, separately for each language task and independently for both the ASD and TD groups, as well as for the differences between the groups.

PLS, akin to Principal Component Analysis (PCA), decomposes the covariance between two data matrices. The first matrix represents brain connectivity, organized as observations multiplied by all possible connections converted into a vector, totaling 124,750 connections, calculated as 500 × (500 − 1)/2. The second matrix represents psychometric assessment, organized as observations multiplied by psychometric variables (n = 44). The covariances are computed across observations. PLS breaks down these covariances into a set of latent variables (LVs) and tests them for significance.

The significance of each LV is evaluated using a permutation test, permuting observations across subjects (conducted 10,000 times in our study). Consequently, each LV is associated with a vector of overall correlations between brain connectivity and psychometrics (with dimensionality equal to the number of psychometric variables, n = 44) and one *p*-value for the entire vector of correlations. To assess the robustness of each connection’s contribution to the identified pattern of correlations, we employed a bootstrap procedure (with 10,000 bootstrap samples). As a result, each connection (out of 124,750) is associated with a bootstrap ratio, equivalent to z-scores, terms we use interchangeably in our study.

For visualization, we convert the vector of z-scores back into a square matrix, organized as 500 Regions of Interest (ROIs) by 500 ROIs. We present the results (including a vector of overall correlations, the corresponding p-value, and a map of z-scores) for the first LV, which accounts for most of the covariance between brain connectivity and psychometric assessments. This analysis elucidates a multivariate pattern of relationships between brain connectivity and psychometric assessment, revealing the average contribution of each psychometric variable and the robustness of individual connections’ contributions to the identified pattern of correlations. 

Z-scores with large magnitudes are indicative of a more robust effect within the analysis. Positive z-scores directly support the overall brain–behavior correlations, while negative z-scores also contribute to the understanding of the contrast, although inversely. It is essential to interpret the map of z-scores and the vector of overall correlations in relation to one another. For instance, a positive z-score for a specific connection, coupled with a positive overall correlation for a particular psychometric variable, signifies a positive correlation. Conversely, a negative z-score for a connection, along with a positive overall correlation for a psychometric variable, implies a negative correlation.

Flipping the vector of overall correlations and z-scores together (i.e., multiplying both correlations and z-scores by −1) does not alter their interpretation. PLS analysis may occasionally flip the output, and to ensure a consistent presentation of our results, we adjusted the overall correlations and z-scores so that the majority of individual correlation values across psychometric variables were positive. This adjustment was applied to maintain the integrity of data interpretation.

## 3. Results

### 3.1. Participants’ Psychometric Profiles 

Significant differences were observed between the ASD and TD groups across various scales, with the exceptions of IQ and the Bystander Cartoon Test, a specific theory of mind (TOM) assessment. These findings affirm that the ASD and TD groups were matched in terms of IQ, providing a controlled comparison. Furthermore, the results indicate that the ASD group exhibited challenges in social communication abilities, encompassing aspects such as social competence and pragmatic language. Table 3, Table 4, Table 5, Table 6, Table 7, Table 8 and Table 9 below present a summary of the descriptive statistics for all the psychometric tests conducted.

### 3.2. Cognitive Performance Associated with Functional Connectivity in TD and ASD Groups

First, we present the results of an analysis that explored correlations between functional brain connectivity and psychometric scores, conducted separately for each group and language task. Using Behavioral Partial Least Squares (PLS) analysis, we uncovered significant correlations between task-dependent connectivity and psychometric scores in both the semantic and pragmatic conditions for both the ASD and TD groups (all *p* < 0.001). While these correlations were present in both groups, the specific patterns of brain–behavior associations exhibited marked differences between the ASD and TD groups. 

The patterns of overall brain–behavior correlations, which are represented by the first latent variable that explains the largest portion of the covariances p for ASD and TD, are presented in Figure 2 and Figure 3, respectively. The corresponding maps of z-scores for ASD and TD are presented in Figure 4 and Figure 5, respectively. Upon visual inspection, we observe consistent patterns of overall brain–behavior correlations in the typically developing (TD) group across both semantic and pragmatic conditions, as depicted in Figure 3. The directionality of these effects—whether correlations are positive or negative—is, on average, maintained across all psychometric measures employed. The robustness of these observed effects is supported in brain space by large in magnitude, positive and negative z-scores, as illustrated in Figure 5. In the context of the prevailing pattern of positive correlations, these z-scores serve as indicators of the corresponding positive and negative brain–behavior relationships, respectively.

The pattern of overall correlations between brain connectivity and psychometric measures in the ASD group diverges qualitatively from that observed in the TD group. Specifically, the directionality of the overall brain–behavior correlations in ASD is inconsistent across different psychometric categories. A subset of variables—namely, those related to IQ (WASI), pragmatic language, and Theory of Mind—exhibits a contrasting pattern compared with the remaining variables. This divergence is consistent across both semantic and pragmatic conditions, as shown in Figure 2. The supporting map of z-scores corresponding to these overall correlations is presented in Figure 4. In the ASD group, higher connectivity between ROI pairs with positive z-scores correlates with elevated scores in the Autism Questionnaire, BRIEF, MSCS, and SRS-2 but lower scores in WASI, pragmatic language, and Theory of Mind. Conversely, higher connectivity between ROI pairs with negative z-scores is associated with lower scores in the Autism Questionnaire, BRIEF, MSCS, and SRS-2 but higher scores in WASI, pragmatic language, and Theory of Mind.

### 3.3. Differences in Correlations between Group Differences in Bran Connectivity and Group Differences in Psychometrics

To explicitly investigate group differences, we conducted an additional PLS analysis alongside the visual examination of brain–behavior correlations for both ASD and TD groups. This analysis aimed to uncover how disparities in psychometric scores between ASD and TD groups could correlate with corresponding differences in brain connectivity. The results are shown in Figure 6 and Figure 7. The observed pattern of overall correlations between the group differences in brain connectivity and psychometric measures was statistically significant (*p* < 0.001, Figure 6). Qualitatively, these patterns closely align with those observed in the ASD group (Figure 3). Moreover, the overall correlations betweengroup differences in brain connectivity and psychometric scores remained consistent across both semantic and pragmatic conditions. Large z-scores, whether positive or negative, indicate the most robust contributions to these overall correlations (Figure 7). Positive z-scores support the overall correlations, signifying negative correlations between brain connectivity and WASI and Theory of Mind variables while showing positive correlations for other psychometric measures. Conversely, negative z-scores can also support the identified overall correlations, but in an inverse manner.

To identify the spatial impact of correlations more precisely between group differences in brain connectivity and psychometric scores, as depicted in the z-score maps (Figure 7), we normalized these maps based on the variability seen in the separate analysis for the ASD group. Specifically, we subtract the z-scores shown in Figure 7 from those in Figure 4, which serve as a baseline. It is important to note that these z-score maps are inherently linked to their respective vectors of overall correlations, making a direct comparison technically inappropriate. However, a visual analysis reveals that the patterns of overall correlations in both cases (Figure 2 and Figure 6) are qualitatively similar. Acknowledging these limitations, we proceed with comparing the corresponding z-score maps.

To carry out baseline normalization, we first normalize the z-score matrices shown in Figure 4 and Figure 7 using L2 normalization. This involves dividing each matrix element by the square root of the sum of squares of all elements within the matrix. This subtraction process is executed separately for both the semantic and pragmatic conditions. Figure 8 displays the z-score maps derived from analyzing correlations between group differences in brain connectivity and psychometric scores (as seen in Figure 7) but now adjusted relative to the baseline set by the z-scores from the ASD-only analysis of brain connectivity and psychometrics (as shown in Figure 4).

As illustrated in Figure 8, the baseline-corrected z-scores are predominantly positive throughout the brain, except for the left and right Limbic A and B networks. It is worth noting that in the original functional atlas, these limbic networks encompass temporopolar and orbital frontal regions, which are susceptible to magnetic resonance (MR) distortions. Due to MR susceptibility effects that spatially distort the signal, the precise boundaries of these networks carry a higher degree of uncertainty compared with other fMRI-identified networks.

Positive baseline-corrected z-scores, indicative of the main effect, suggest that the brain’s spatial functional organization is in direct alignment with the overall correlation patterns depicted in Figure 6. In more precise terms, for most network pairings, an increase in group differences in functional connectivity correlates with an increase in group differences in scores for the Autism Questionnaire, BRIEF, MSCS, and SRS-2. Conversely, it correlates with a decrease in group differences for WASI, pragmatic language, and Theory of Mind scores.

## 4. Discussion

The present study investigated associations between task-dependent functional connectivity during semantic and pragmatic language processing and relevant cognitive abilities in children with ASD and their typically developing peers. We have found that the networks engaged during the language tasks were associated with performance on psychometric tests of abilities in relevant cognitive domains. Notably, this relationship shows distinctions between the ASD and TD groups, especially concerning IQ, pragmatic language from CASL-2 assessment, and theory of mind (TOM). The different patterns of brain–behavior connections observed in the ASD and TD groups prompt questions about whether these differences reflect compensatory and adaptive brain network recruitment in ASD, inefficiency in selectively integrating the necessary neuronal groups for task performance among individuals with ASD, or simply neurologically distinct processing of the same information in the brains of those with ASD. 

While numerous studies have delved into atypical brain–behavior relationships in ASD, our research is novel in multiple respects. While prior work has predominantly focused on distinguishing ASD and TD characteristics through resting-state fMRI brain connectivity patterns [77,78,79] with connectivity estimated during cognitive tasks [20,80,81,82,83], our study occupies a unique niche. Our first set of analyses emphasizes associations between psychometrics and pragmatic and semantic language processing. The second set directly contrasts differences in brain–behavior associations between the ASD and TD groups.

Unlike previous studies that largely emphasized the disparities in within-group relationships between brain connectivity and psychometrics [20,80,81,82,83], our study pioneers an examination of inter-group relationships by correlating the differences in brain connectivity with differences in behavior between ASD and TD. This method offers a direct insight into whether discrepancies in brain and behavior between the two groups are related or operate independently. Our study also takes the novel approach of gauging task-dependent connectivity during semantic and pragmatic conditions, which was not explored previously. 

More specifically, in our first analysis investigating associations between psychometric scores and brain network connectivity, we observed a common trend in both ASD and TD participants, where superior performance was associated with higher connectivity, except for IQ, pragmatic language, and TOM scores in the ASD group. This finding may indicate that the networks that people with ASD recruited might be maladaptive. Alternatively stated, the more they rely on these networks, the poorer their performance becomes. Simplified further, it suggests that they may be engaging in task-irrelevant connections, which is associated with poorer cognitive performance in pertinent domains. It is important to note, however, that the ASD and TD groups did not differ in their performance on IQ and TOM tests. In fact, IQ and TOM are the only two tasks in which ASD and TD do not differ significantly, which indicates that the ASD group did not perform worse because of how they differ from their neurotypical peers in terms of brain–behavior relationships. It is actually not uncommon that individuals with autistic traits have normal IQ and cognitive performance compared with typically developing individuals despite atypical connectivity patterns [10,84]. This suggests that network recruitment in ASD may not simply be maladaptive and that the networks they recruit during language tasks are associated with better performance in relevant cognitive domains. 

In our second analysis, we examined the relationship between differences in connectivity and differences in behavior between the ASD and TD groups. Our findings indicate that the more distinct the connectivity between these groups, the greater the differences in cognitive performance, except for the IQ and TOM tasks. The results regarding the pragmatic language task were inconclusive. Specifically, for the IQ and TOM tasks, the greater the disparity in connectivity and network recruitment between the ASD and TD groups, the closer the performance of the ASD group approached that of the TD group. This suggests that network recruitment in ASD is not maladaptive but rather compensatory. In this sense, children with ASD may be achieving comparable cognitive performance in these relevant areas by using different patterns of brain network recruitment. This is consistent with other research in which ASD showed increased connectivity in networks that were not employed in their cognitively matched peers during the performance of cognitive tasks, in which authors also suggested the compensatory nature of networks recruited in ASD [84,85,86,87]. 

Previous studies have provided evidence that individuals with ASD exhibit atypical structural and functional brain characteristics. These include reports of brain volume overgrowth, particularly in the frontal and temporal lobes [88], as well as reversed asymmetry in typically lateralized language areas [89]. Additionally, there is evidence indicating that other neurodevelopmental disorders, such as Attention-Deficit Hyperactive Disorder (ADHD) and very premature birth, also involve altered brain structure and function [90,91]. By drawing parallels between these different conditions, we can gain a better understanding of each of them, with a particular focus on ASD for the purpose of this study. In ASD, ADHD, and preterm-born adolescents, we can often observe more connectivity, which occurs in addition to that seen in their typically developing peers during task-dependent fMRI recording. In ASD and preterm adolescents, one example is the increased right hemisphere involvement in language tasks compared with controls [4,91]. In ADHD, it is the increased involvement of the right prefrontal regions, left dorsal cingulate cortex, and left cuneus during a working memory task compared with controls [92]. In many of these cases, the performance of people with neurodevelopmental difficulties did not differ significantly from that of controls [4,91,92]. Some researchers have suggested that these atypical and “extra” brain networks that ADHD or preterm-born adolescents recruit might be indicative of the adaptation of neural networks to the altered structural properties of the brain and that they may serve a compensatory role. 

Given prior findings in ASD and other pediatric populations and the results from the present study, we believe that ASD brain network recruitment could be compensatory in nature. The atypical networks that ASD adopts support cognitive performance when the typical networks might not be sufficient. Hence, the more the networks of ASD differ from TD, the closer they perform on the level of TD. This can be reflected in our study, with IQ and TOM tasks being the only two tasks that ASD and TD do not differ in performance. 

Although there seem to be adaptive mechanisms in people with neurodevelopmental disorders, the resulting compensatory network connectivity might not function as efficiently as typical connectivity. For example, graph-based network analysis on ADHD children has suggested that their networks are less efficient than controls [93]. This could potentially explain the negative correlation between connectivity and task performance in our study for TOM, IQ, and pragmatic language tasks from CASL-2 assessment in ASD in our first set of analyses. The inefficiency of compensatory network recruitment is also suggestive that there could be conditions where individuals with neurodevelopmental disorders may not be able to adequately compensate. This could be when the tasks are too difficult or when the tasks rely on quick processing speed. We believe that this might account for the pattern of effects observed for the CASL-2 pragmatic language task in our study, where associations between brain connectivity and cognitive abilities in the ASD group significantly differ from that observed for the TD group. The CASL-2 pragmatic language task does not have a clear direction of correlation in our second set of analyses: an increase in the difference in connectivity between ASD and TD can be related to an increase or decrease in the difference in task performance. We believe that this relationship can be indicative of how the compensatory networks fail to compensate for potential deficits in networks that are responsible for pragmatic language function in ASD. 

The development and existence of such dynamic adaptation and reorganization of neural connectivity underscores the importance of studying neurodevelopmental disorders with different age groups and properly accounting for age and brain maturity during the study. For example, adolescents and adults with ASD may demonstrate different levels of functional connectivity alterations during a language task compared with controls of the same age group [94]. This could be because the compensatory networks are at different stages of development, which would also explain how ASD individuals of different age groups may have different levels of adaptation to their social environments. This finding has implications for future research directions. We believe that an investigation into how ASD individuals from different age groups differ from controls from the same age group in terms of functional connectivity and brain–behavior relationships would be beneficial to understanding ASD and neurodevelopmental disorders in general. We also believe that studies that look into the physical evidence of compensatory networks would be helpful. For example, by looking into white matter tracts using diffusion tensor imaging, we might attain a better understanding of how the brain adapts and organizes itself and brain plasticity in general. 

## 5. Limitations

We acknowledge that our study has certain limitations, notably the small sample size. However, even with this constraint, our study offers valuable insights into the functional connectivity patterns in individuals with ASD. We specifically focus on task-dependent functional connectivity during pragmatic and semantic language tasks and explore their correlation with cognitive performance in these areas.

## 6. Conclusions

This study is one of the first to look into associations between task-dependent functional connectivity and cognitive abilities in relevant domains in terms of how they differ between TD and ASD groups. We have found different brain–behavior relationships between ASD and TD, which could be indicative of adaptive networks that people with ASD develop to compensate for potential functional deficits or distinct processing of the same information. We also discussed the potential limitations of such compensatory networks, such as their inefficiency. Studies of functional brain connectivity in ASD, taking into account age and the development trajectories of the compensatory networks, as well as their association with structural properties of the brain, present themselves as appealing avenues for further investigation. 

## Figures and Tables

**Figure 1 brainsci-13-01448-f001:**
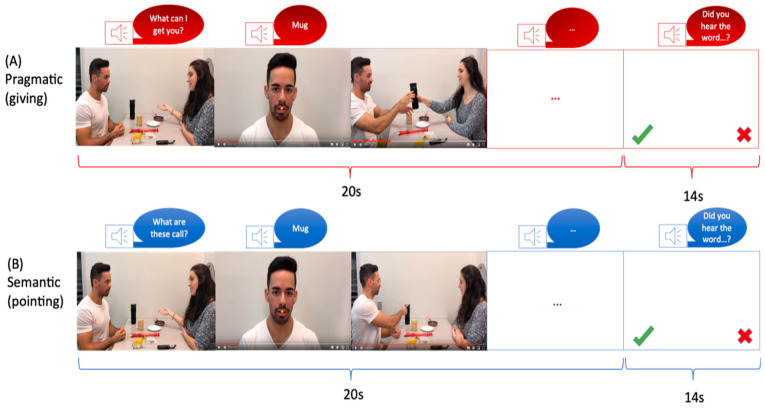
Schematic illustration of fMRI stimuli for (**A**) semantic condition and (**B**) pragmatic condition. Note. A trial sequence started with a display of objects and communicating actors. A context sentence (e.g., “What can I get you?” in the pragmatic condition (**A**) or “What are these called?” in the semantic condition (**B**), was uttered by the Partner. Following this, a series of five scenes was shown, in which the Speaker’s face appeared together with the critical spoken utterance, which served for naming (semantic) vs. requesting (pragmatic). The words were identical for both speech acts. The word scenes were followed by a series of five acting scenes involving the objects mentioned in the worked utterances (handing over an object in the requesting condition or pointing at it in the naming condition).

**Figure 2 brainsci-13-01448-f002:**
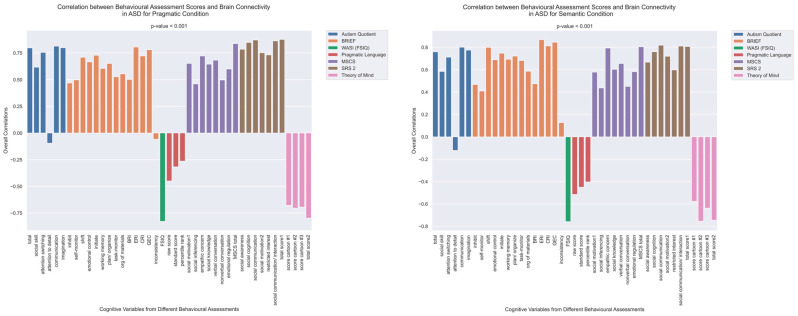
The patterns of overall correlations between functional connectivity and psychometrics in ASD, as revealed by Behavioral PLS analysis. One psychometric score is associated with one correlation value, which was estimated in a multivariate way across all the fMRI features (functional connectivity for all ROI pairings) using Behavioral PLS analysis. Variables “score cartoon #1”, “score cartoon #2”, and “score cartoon #3” represents scores from the three scenarios in the Bystander Cartoon tests (Theory of Mind).

**Figure 3 brainsci-13-01448-f003:**
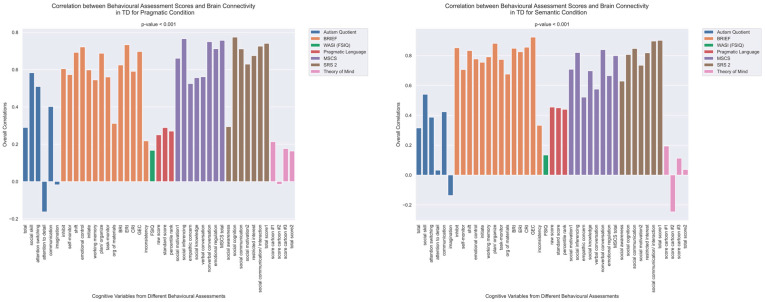
The patterns of overall correlations between psychometrics and brain connectivity in TD, as revealed by Behavioral PLS analysis. Variables “score cartoon #1”, “score cartoon #2”, and “score cartoon #3” represents scores from the three scenarios in the Bystander Cartoon tests (Theory of Mind).

**Figure 4 brainsci-13-01448-f004:**
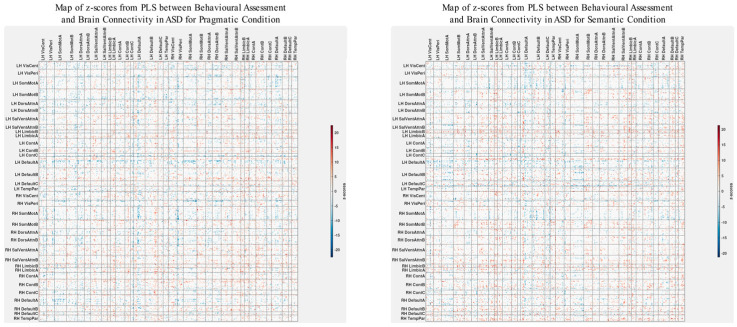
The maps of z-scores illustrate the robustness of contributions of individual connections (ROI pairings) to the overall correlations in TD, as shown in Figure 2. The maps are shown separately for the pragmatic and semantic conditions. Displayed as 500 × 500 matrices, these maps correspond to the 500 Regions of Interest (ROIs) defined in the parcellation atlas. Each ROI is allocated to one of 17 functional MRI networks for each hemisphere (LH and RH).

**Figure 5 brainsci-13-01448-f005:**
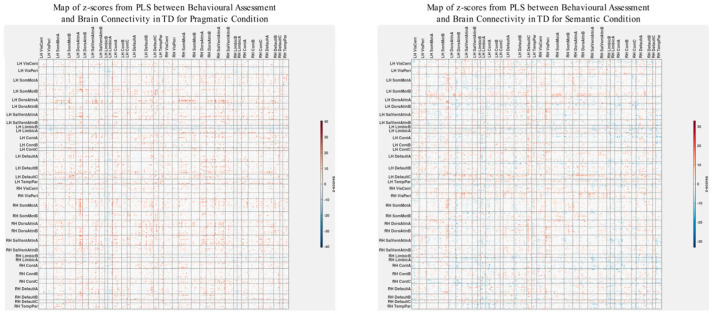
The maps of z-scores illustrate the robustness of contributions of individual connections (ROI pairings) to the overall correlations in TD, as shown in Figure 3. Similarly to Figure 4, these maps correspond to the 500 Regions of Interest (ROIs) defined in the original parcellation of the brain.

**Figure 6 brainsci-13-01448-f006:**
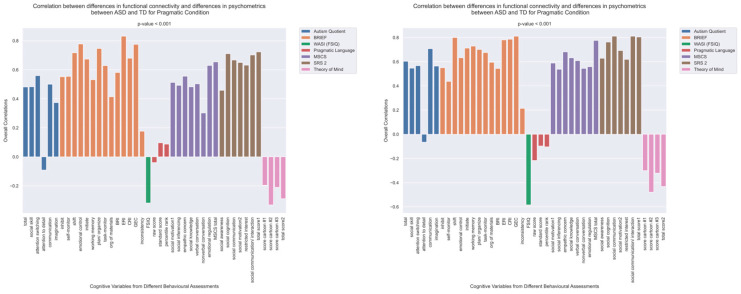
The patterns of overall correlations between group differences in bran connectivity and group differences in psychometrics. These patterns are shown separately for the pragmatic and sematic conditions. Variables “score cartoon #1”, “score cartoon #2”, and “score cartoon #3” represents scores from the three scenarios in the Bystander Cartoon tests (Theory of Mind).

**Figure 7 brainsci-13-01448-f007:**
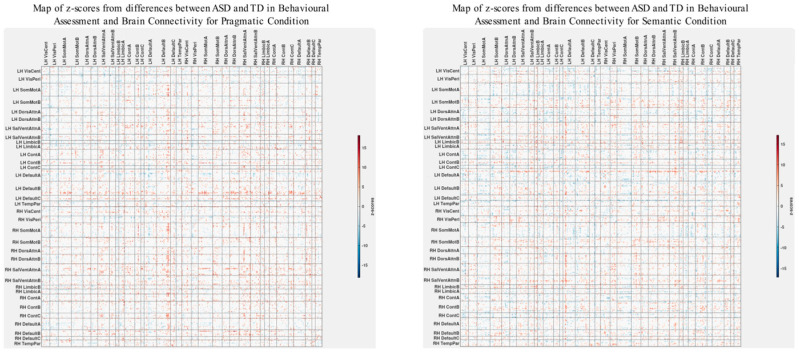
Maps of z-scores from PLS analysis exploring correlations (Figure 6) between group differences in psychometric scores and group differences in brain connectivity.

**Figure 8 brainsci-13-01448-f008:**
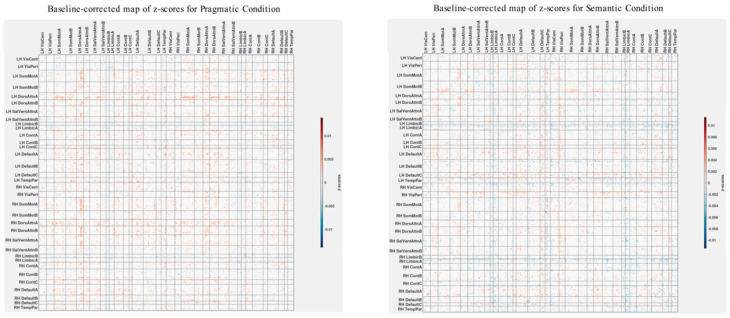
Baseline-corrected maps of z-scores associated with the analysis of correlations between group differences in brain connectivity and group differences in psychometric scores for the pragmatic and semantic conditions (Figure 6 and Figure 7).

**Table 1 brainsci-13-01448-t001:** Sample Statistics and AQ Questionnaire and FSIQ 2 Test Result.

	Children without ASD (*n* = 19)	Children with ASD (*n* = 16)	*t* Value	*p*	Effect Size
Age (years)					
***M*** (*SD*)	9.55 (1.56)	10.15 (1.32)	−1.22	0.23	0.42
Range	7.20–12.90	4.60–12.00			
IQ					
***M*** (*SD*)	110.84 (11.44)	102.75 (19.81)	−7.86	0.16	0.50
Range	82–130	62–137			
Gender					
Male	13	13	−0.85	0.40	0.43
Female	6	3			
AQ total					
***M*** (*SD*)	13.00 (6.94)	32.63 (7.83)	−7.86	*p* < 0.01	2.65
Range	2–26	14–42			

Note: *M* = mean, *SD* = standard deviation, AQ = autism spectrum quotient, IQ = intelligence quotient.

**Table 2 brainsci-13-01448-t002:** Psychometric Outcome Measures (Cognitive-Behavioral Assessments).

Domain	Test	Subtest
Executive functions	Behavior Rating Inventory of Executive Function (BRIEF-2)	All
Behavior mental health	Behavior Assessment System for Children (BASC-2): Parents questionnaire	All
Cognitive ability	Wechsler Abbreviated Scale of Intelligence (WASI-II)	Intelligence Quotient (IQ)
Social competence	Chandler Cartoons Questionnaire	Sandcastle, Airplane, and Snowman cartoons
	Social Style Questionnaire (4 to 12 years old)	All
	Social Responsiveness Scale (SRS 2)	All
	Multidimensional Social Competence Scale (MSCS)	All
Pragmatic language	Comprehensive Assessment of Spoken Language (CASL 2)	Pragmatic languages

Note. Demographic Questionnaire, Behavior Rating Inventory of Executive Function (BRIEF-2), Behavior Assessment System for Children (BASC-2), Wechsler Abbreviated Scale of Intelligence (WASI-II), Chandler Cartoons Questionnaire, Social Style Questionnaire (4 to 12 years old), Social Responsiveness Scale (SRS 2), Multidimensional Social Competence Scale (MSCS), Comprehensive Assessment of Spoken Language (CASL 2).

**Table 3 brainsci-13-01448-t003:** Descriptive Statistics for BRIEF-2.

Scales (*t* Scores)	Children without ASD (*n* = 19)	Children with ASD (*n =* 16)	*t* Value	*p*	Effect Size
BRI					
*M* (*SD*)	16.58 (5.04)	26.94 (5.31)	−5.91	*p* < 0.01	2.00
Range	12–29.0	17–34			
Inhibit					
*M* (*SD*)	11.53 (3.64)	17.63 (3.88)	−4.76	*p* < 0.01	1.62
Range	8–20.0	10–22.0			
Self-monitor					
*M* (*SD*)	5.05 (1.75)	9.31 (1.70)	−7.29	*p* < 0.01	2.47
Range	4–9.0	7–12.0			
ERI					
*M* (*SD*)	22.68 (7.21)	37.00 (7.29)	−5.82	*p* < 0.01	1.97
Range	16–42	21–47			
Shift					
*M* (*SD*)	11.32 (3.00)	18.88 (3.76)	−6.62	*p* < 0.01	2.22
Range	8–19.0	12–24.0			
Emotional control					
*M* (*SD*)	11.37 (4.45)	18.13 (4.84)	−4.30	*p* < 0.01	1.45
Range	8–23.0	9–24.0			
CRI					
*M* (*SD*)	50.11 (12.75)	74.19 (14.73)	−5.19	*p* < 0.01	1.75
Range	33–77	47–94			
Initiate					
*M* (*SD*)	7.16 (1.74)	11.75 (2.35)	−6.46	*p* < 0.01	2.22
Range	5–11.0	8–15.0			
Working memory					
*M* (*SD*)	11.42 (3.19)	17.69 (4.45)	−4.84	*p* < 0.01	1.62
Range	8–18.0	8–23.0			
Plan/organize					
*M* (*SD*)	12.68 (3.45)	19.75 (3.30)	−6.16	*p* < 0.01	2.10
Range	8–20.0	14–24			
Task-monitor					
*M* (*SD*)	8.58 (2.71)	12.38 (3.01)	−3.92	*p* < 0.01	1.33
Range	5–13.0	7–15.0			
Organization of materials					
*M* (*SD*)	10.26 (3.02)	12.63 (3.69)	−2.09	0.45	0.7
Range	6–18.0	7–18.0			
GEC					
*M* (*SD*)	89.37 (22.96)	138.13 (24.86)	−6.03	*p* < 0.01	2.04
Range	61–144	91–169			

**Table 4 brainsci-13-01448-t004:** Descriptive Statistics for Autism Quotient Questionnaire.

Scales (*t* Scores)	Children without ASD (*n* = 19)	Children with ASD (*n =* 16)	*t* Value	*p*	Effect Size
Social skill					
*M* (*SD*)	1.74 (1.91)	5.81 (2.46)	−5.52	*p* < 0.01	1.85
Range	0–8	1–10.0			
Attention switching					
*M* (*SD*)	3.21 (2.18)	7.13 (2.53)	−4.93	*p* < 0.01	1.66
Range	0–7	2–10.0			
Attention to detail					
*M* (*SD*)	4.53 (2.88)	6.56 (2.10)	−2.35	0.03	0.81
Range	0–10	2–10.0			
Communication					
*M* (*SD*)	1.58 (1.35)	7.00 (2.42)	−7.98	*p* < 0.01	2.77
Range	0–5	1–10.0			
Imagination					
*M* (*SD*)	1.95 (1.03)	6.13 (2.06)	−7.37	*p* < 0.01	2.57
Range	0–4	2.0–9			
AQ total					
*M* (*SD*)	13.00 (6.94)	32.63 (7.83)	−7.86	*p* < 0.01	2.65
Range	2.0–26	14–42			

**Table 5 brainsci-13-01448-t005:** Descriptive Statistics for MSCS.

Scales (*t* Scores)	Children without ASD (*n* = 19)	Children with ASD (*n =* 16)	*t* Value	*p*	Effect Size
Social motivation					
*M* (*SD*)	43.05 (7.47)	32.56 (10.47)	3.45	*p* < 0.01	1.15
Range	28–55	11–52			
Social inferencing					
*M* (*SD*)	44.68 (6.46)	24.63 (8.35)	8.01	*p* < 0.01	2.69
Range	29–53	15–49			
Empathic concern					
*M* (*SD*)	46.00 (6.88)	30.69 (8.74)	5.80	*p* < 0.01	1.95
Range	31–55	16–54			
Social knowledge					
*M* (*SD*)	47.53 (6.70)	35.25 (8.20)	4.88	*p* < 0.01	1.64
Range	30–54	19–48			
Verbal conversation					
*M* (*SD*)	41.68 (8.38)	27.06 (9.70)	4.79	*p* < 0.01	1.61
Range	22–53	13–49			
Nonverbal conversation					
*M* (*SD*)	49.95 (4.62)	37.75 (10.20)	4.42	*p* < 0.01	1.54
Range	39–55	19–53			
Emotional regulation					
*M* (*SD*)	41.58 (8.09)	27.69 (9.24)	4.74	*p* < 0.01	1.60
Range	22–51	13–43			
MSCS total					
*M* (*SD*)	314.47 (39.99)	215.63 (46.36)	6.78	*p* < 0.01	2.28
Range	234–364	134–329			

**Table 6 brainsci-13-01448-t006:** Descriptive Statistics for SRS-2.

Scales (*t* Scores)	Children without ASD (*n* = 19)	Children with ASD (*n =* 16)	*t* Value	*p*	Effect Size
Social awareness					
*M* (*SD*)	45.11 (9.68)	67.69 (13.27)	−5.81	*p* < 0.01	1.94
Range	34–70	35–89			
Social cognition					
*M* (*SD*)	45.21 (7.38)	73.00 (10.55)	−8.87	*p* < 0.01	3.05
Range	39–63	55–90			
Social communication					
*M* (*SD*)	45.89 (7.56)	73.00 (11.90)	−8.17	*p* < 0.01	2.72
Range	38–65	44–90			
Social motivation					
*M* (*SD*)	49.16 (10.29)	68.75 (12.48)	−5.09	*p* < 0.01	1.71
Range	38–75	46–90			
Restricted interests/repetitiveness					
*M* (*SD*)	49.00 (8.52)	74.75 (12.50)	−7.22	*p* < 0.01	2.41
Range	41–71	50–90			
Social communication/interaction					
*M* (*SD*)	46.05 (8.22)	73.56 (11.29)	−8.33	*p* < 0.01	2.79
Range	38–66	46–90			
SRS-2 total					
*M* (*SD*)	46.37 (8.24)	74.50 (11.02)	−8.63	*p* < 0.01	2.89
Range	38–67	47–90			

**Table 7 brainsci-13-01448-t007:** Descriptive Statistics for CASL-2.

Scales (*t* Scores)	Children without ASD (*n* = 19)	Children with ASD (*n =* 16)	*t* Value	*p*	Effect Size
CASL-2 Pragmatic language					
*M* (*SD*)	109.84 (14.98)	90.19 (13.32)	4.07	*p* < 0.01	1.38
Range	81–141	70–117			

**Table 8 brainsci-13-01448-t008:** Descriptive Statistics for IQ.

Scales (*t* Scores)	Children without ASD (*n* = 19)	Children with ASD (*n =* 16)	*t* Value	*p*	Effect Size
WASI-2 IQ					
*M* (*SD*)	110.84 (11.44)	102.75 (19.81)	−7.86	0.16	0.5
Range	82–130	62–137			

**Table 9 brainsci-13-01448-t009:** Descriptive Statistics for Bystander Cartoon Test (Theory of Mind).

Scales (*t* Scores)	Children without ASD (*n* = 19)	Children with ASD (*n =* 16)	*t* Value	*p*	Effect Size
Bystander cartoon total					
*M* (*SD*)	9.00 (4.06)	6.31 (4.63)	1.83	0.08	0.62
Range	0–12				

Note: Bystander cartoon descriptive statistics.

## Data Availability

Data available upon request.

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
