# Peer review of "Atypical Associations between Functional Connectivity during Pragmatic and Semantic Language Processing and Cognitive Abilities in Children with Autism"

_brainsci, 2023, doi:10.3390/brainsci13101448_

Round 1

Reviewer 1 Report

Dear Authors,

please correct minor language errors. The article has its value, but it can be enriched by citing the following works: PMID: 35377335 DOI: 10.26402/jpp.2021.6.01 and PMCID: PMC9334648 DOI: 10.3389/fpsyt.2022.942218. It is also worth adding a separate clear paragraph regarding the limitations of the study and conclusion.

Good luck!

Author Response

please correct minor language errors. The article has its value, but it can be enriched by citing the following works: PMID: 35377335 DOI: 10.26402/jpp.2021.6.01 and PMCID: PMC9334648 DOI: 10.3389/fpsyt.2022.942218. It is also worth adding a separate clear paragraph regarding the limitations of the study and conclusion.

Good luck!

Thank you for your suggestions.  We have incorporated these references into the Introduction, Assessments, and Conclusions sections.  We have also added a section on limitations and conclusion.

Reviewer 2 Report

Thank you for the opportunity to review this interesting topic. My recommendations regarding the current manuscript are:

In the Introduction section after the presentation of the objectives of the study, the authors should present the hypotheses of every presented aim. What are the hypotheses and the literature that support them? Also in the Introduction, the authors must explain the innovation of the current study as well as its significance. Is there any research gap that this study attempts to enlighten?

In the methodology, the authors present information that must be presented in measurements. The authors must add a Measurement section, and there they have to present all the assessment tools that applied. For example, the authors mention the cognitive abilities evaluation. This is very unclear. The evaluated memory, attention, and perception? Also the Procedure section is missing. The authors must describe the procedure of the evaluation, and if there is ethical aprooval of the study.

The Discussion is not solid. The authors must present new research studies to explain their research findings. Furthermore, there is lack of used references in the Discussion. With only 8 references the authors can not interpet their findings. 

Finally, the authors must explain the implications of the current study. 

Author Response

Thank you for the opportunity to review this interesting topic. My recommendations regarding the current manuscript are:

In the Introduction section after the presentation of the objectives of the study, the authors should present the hypotheses of every presented aim. What are the hypotheses and the literature that support them? Also in the Introduction, the authors must explain the innovation of the current study as well as its significance. Is there any research gap that this study attempts to enlighten?

Thank you for your  suggestions.  We have clearly highlighted the hypotheses and  knowledge gap (page 3). 

In the methodology, the authors present information that must be presented in measurements. The authors must add a Measurement section, and there they have to present all the assessment tools that applied. For example, the authors mention the cognitive abilities evaluation. This is very unclear. The evaluated memory, attention, and perception? Also the Procedure section is missing. The authors must describe the procedure of the evaluation, and if there is ethical approval of the study.

Our measurements are detailed in the “Behavioural Assessments” section for cognitive measurements and “MRI Data Acquisition” section for brain measurements in the revised manuscript.  Table 2 presents a list of the specific tests that were used in our study to measure performance in different domains like executive functions, social competence etc. 

Ethical approval is mentioned in the last line of the first paragraph of “Participants” section of the revised manuscript.

The Discussion is not solid. The authors must present new research studies to explain their research findings. Furthermore, there is lack of used references in the Discussion. With only 8 references the authors can not interpret their findings. 

Finally, the authors must explain the implications of the current study. 

We have incorporated more references in the Discussion section to explain our findings in the context of current literature.  We also discussed the implications of the study within such context in terms of future research directions in the conclusions section. 

Reviewer 3 Report

The paper proposes an important study on differences in brain activity and connectivity in ASD and TD children engaged in semantic and pragmatic language processing tasks.

Results are convincingly discussed, though a small experimental group. It must be acknowledged that the authors explicitly write that the sample size is small.

Results have, however, a great impact, since they suggest that ASD subjects are characterised by a brain-behaviour connectivity that differs from TD peers, though producing IQ and teory of mind-based tests that do not differ significantly. The evidence is interpreted as an adaptively compensatory brain network, thus resulting in comparable cognitive performance. 

An additional PRO of the paper relies in the inclusion in the discussion section of results related to ADHD studies. 

I habe only a few suggestions for improve it.

z-scores matrixes do not clearly dysplay results (maybe due to the color code).

As a general stylistic comment, there are tons of references not included in the list, especially those referred to the protocols (e.g. AQ questionnaire by Baron-Cohen, the BRIEF-2, BASC3, CASL2, etc.), as well as the reference to a paper of yours submitted. In the latter case, either you add the full reference or you delete it.

Abstract: .., children with ASD and their typically developing (TD) peers ... was mapped --> were compared.

Introduction (last paragraph):  ... differences observed in ASD and TD groups in terms of relations brain-behaviour relantionships --> one relation to erase

Author Response

The paper proposes an important study on differences in brain activity and connectivity in ASD and TD children engaged in semantic and pragmatic language processing tasks.

Results are convincingly discussed, though a small experimental group. It must be acknowledged that the authors explicitly write that the sample size is small.

Results have, however, a great impact, since they suggest that ASD subjects are characterised by a brain-behaviour connectivity that differs from TD peers, though producing IQ and theory of mind-based tests that do not differ significantly. The evidence is interpreted as an adaptively compensatory brain network, thus resulting in comparable cognitive performance. 

An additional PRO of the paper relies in the inclusion in the discussion section of results related to ADHD studies. 

I have only a few suggestions for improve it.

z-scores matrixes do not clearly display results (maybe due to the color code).

As a general stylistic comment, there are tons of references not included in the list, especially those referred to the protocols (e.g. AQ questionnaire by Baron-Cohen, the BRIEF-2, BASC3, CASL2, etc.), as well as the reference to a paper of yours submitted. In the latter case, either you add the full reference or you delete it.

Abstract: .., children with ASD and their typically developing (TD) peers ... was mapped --> were compared.

Introduction (last paragraph):  ... differences observed in ASD and TD groups in terms of relations brain-behaviour relationships --> one relation to erase

Thank you for the feedback.  We have properly revised the bibliography.  We have also made other changes, as suggested.

 We have chosen the color scheme for the z-score maps to enhance intuitive interpretation. We aimed to present comprehensive data from the analysis while ensuring clarity. We employed a multivariate PLS analysis, and our focus was on the overall correlation patterns, which were properly descried in the Results section. For readers keen on a granular, region-by-region analysis, we have made available the maps of z-scores, high-resolution images that can be downloaded and closely examined.

Round 2

Reviewer 2 Report

The manuscript is very intersting. The content of the manuscript is very solid.